# Simulation of the seasonal and spatial variability of the concentrations and chemical composition of ultrafine particulate matter over Europe

Konstantinos Mataras[1*], Evangelia Siouti[2*], David Patoulias[2] and Spyros N. Pandis[1,2]

[1]Department of Chemical Engineering, University of Patras, Patras, Greece

[2]Institute of Chemical Engineering Sciences (ICE-HT), Foundation for Research and Technology Hellas (FORTH), Patras, Greece

*These two authors contributed equally to this work.

*Correspondence to*: Spyros N. Pandis (spyros@chemeng.upatras.gr)

**Abstract.** Ultrafine particles (UFPs) have attracted interest as perhaps the most dangerous fraction of atmospheric PM. This study focuses on the simulation of ultrafine particulate matter ($PM_{0.1}$) mass concentrations and their chemical composition during a summer and winter period in Europe.

Predicted levels of $PM_{0.1}$ varied substantially, both in space and in time. The average predicted $PM_{0.1}$ mass concentration was 0.6 μg m$^{-3}$ in the summer, higher than the 0.3 μg m$^{-3}$ predicted in the winter period. $PM_{0.1}$ chemical composition exhibited significant seasonality. In summer, $PM_{0.1}$ was mostly comprised of secondary inorganic matter (38% sulfate and 13% ammonium) and organics (9% primary and 32% secondary). During the winter, the fraction of secondary inorganic matter increased, with sulfate contributing 47% and ammonium 19%, on average. Primary organic matter contribution also increased from 9% in summer to 23% in winter, while secondary organic matter decreased significantly to 6% on average during winter.

During summertime, the model performance at 12 sites for daily average ultrafine particle volume ($PV_{0.1}$) concentrations was considered good, with normalized mean error (NME) equal to 46% and normalized mean bias (NMB) equal to 15%. For the winter period, the corresponding values for daily average levels were -27% for NMB and 64% for NME, indicating an average model performance.

Correlations between $PM_{0.1}$ and the currently regulated $PM_{2.5}$ (particulate matter with a diameter lower than 2.5 μm) were generally low. Better correlations were observed in cases where the primary component of $PM_{0.1}$ was significant. This suggests that there are significant differences between the dominant sources and processes of $PM_{0.1}$ and $PM_{2.5}$.

## 1. Introduction

UFPs dominate atmospheric particle number distribution (Seinfeld and Pandis, 2006). High concentrations of both UFP number and mass are found in urban areas and are a result of human activity, directly emitting particulates or producing them by gas-to-particle conversion processes. Atmospheric particle exposure is one of the most significant risk factors affecting human health (HEI, 2013; EPA, 2019). Ultrafine particles have attracted interest because they may be the most dangerous fraction of atmospheric particulate matter. They can reach the lung alveoli, pass into the bloodstream and from there they can move to many different organs (Schraufnagel, 2020; Sioutas et al., 2005). Their increased specific surface area (total surface area of the particles per unit mass) with decreasing size also enhances their chemical and physical interactions, both with gaseous species outside the body and also with tissues inside the body (Kwon et al., 2020). Some

epidemiological studies have noted a positive correlation between UFPs exposure and brain tumor incidence (Weichenthal et al., 2020). However, there are still questions about the links between ultrafine particle exposure and damage to human health (EPA, 2019).

Past studies of ultrafine particles have focused on their number concentrations (Baranizadeh et al., 2016; Merikanto et al., 2009; Patoulias et al., 2015, 2018; Wang and Penner, 2009; Yu and Luo, 2009). The comparatively scarce modelling attempts aimed at ultrafine particle mass have mostly been conducted in California and the US (Hu et al., 2014a, b, 2017; Venecek et al., 2019; Yu et al., 2019).

In the study by Hu et al. (2014a, b) for the 7-year (2000-2006) period, daily predictions of primary $PM_{0.1}$ from the UCD-P (University of California Davis-Primary) model were evaluated for California. They found good agreement of model predictions with observed $PM_{0.1}$ mass and elemental carbon (EC), with a Pearson correlation coefficient (R>0.92) during these periods (Kuwayama et al., 2013). They reported model difficulties in reproducing observed values of $PM_{0.1} > 4$ µg m$^{-3}$ or $< 1$ µg m$^{-3}$. In a subsequent study of $PM_{0.1}$, Hu et al. (2017) utilized again the UCD/CIT (University of California Davis/California Institute of Technology) model. The authors reported that primary organic matter was the major component (50-90%) of $PM_{0.1}$ organic aerosol (OA) in California, with 9-year average concentrations above 2 µg m$^{-3}$ in major urban areas. They predicted that secondary organics contribute less than 10% to $PM_{0.1}$ OA in these areas, with that contribution increasing to up to 50% in rural areas, with low organic matter content. $PM_{0.1}$ secondary organic aerosol (SOA) concentrations were predicted to be mostly biogenic (64% of SOA for the domain) and between 0.02-0.05 µg m$^{-3}$ in the winter and 0.1-0.3 µg m$^{-3}$ in the summer. Underprediction of secondary organic aerosol concentrations was proposed as an explanation of the $PM_{0.1}$ organic mass underprediction. Yu et al. (2019) along with Venecek et al. (2019) considered nucleation along with the rest of the major aerosol processes in a $PM_{0.1}$ study. Venecek et al. (2019) investigated $PM_{0.1}$ concentration and sources during summertime pollution events in several metropolitan areas of the US. Predicted daily average $PM_{0.1}$ levels were generally above 2 µg m$^{-3}$, reaching 5 µg m$^{-3}$ in areas influenced by wildfire events. The $PM_{0.1}$ spatial gradients were much sharper than those of $PM_{2.5}$ due to the dominance of the primary $PM_{0.1}$. The dominant source of $PM_{0.1}$ was found to be natural gas combustion across all major cities in the US. Yu et al. (2019) studied UFP number as well as mass concentrations and sources in California. Xue et al. (2019) reported that meat cooking was a major source of $PM_{0.1}$ organic carbon across all California cities (13−29%), while nucleation contributed negligibly to UFP mass on an annual scale.

Experimental studies investigating ultrafine particles have focused on particle number concentrations and their spatial and temporal differences. The first detailed measurements of UFP mass have been performed in California (Kuwayama et al., 2013; Xue et al., 2018, 2019, 2020a, b; Xue and Kleeman, 2022). In these studies, researchers collected one sample every day or used even longer averaging intervals because of the low UFP mass concentrations. Hughes et al. (1998) reported daily average mass concentrations varying from 0.8 to 1.6 µg m$^{-3}$ in Pasadena, CA. A novel method to measure UFP mass continuously has been recently developed and tested by Argyropoulou et al. (2023, 2024), but has not been applied in field studies yet.

Major sources of $PM_{0.1}$ in the US include vehicular emissions (Hu et al., 2014a), biomass (wood burning and meat cooking) burning (Kleeman et al., 2009) but also natural gas combustion (Xue et al., 2018) and aviation in areas close to airports (Venecek et al., 2019). Relatively little is known in areas outside the US about ultrafine particle mass properties other than their number concentrations and size distribution (del Águila et al., 2018; Putaud et al., 2010).

The few studies, however, using $PM_{0.1}$ as the exposure metric have shown positive correlations of ultrafine
particle organic and trace metal components with negative health effects (Laurent et al., 2016; Ostro et al., 2015). For
UFP mass, field studies as well as modelling studies have been largely restricted to California or parts of Asia, which are
dominated by primary sources (Phairuang et al., 2022; Xue et al., 2019, 2020b; Zhu et al., 2002). As such, large
uncertainties about their health effects still remain (Delfino et al., 2005; EPA, 2019; Ohlwein et al., 2019).
In this work, $PM_{0.1}$ mass concentrations as well as their chemical composition were studied during a typical
summer (5 June - 8 July 2012) and winter period (1-30 January 2009) for several urban and rural sites in Europe using
the PMCAMx-UF (Particulate Matter Comprehensive Air-quality Model with extensions – Ultra-Fine) chemical transport
model (CTM). Due to the difficulty of measuring $PM_{0.1}$ mass, $PV_{0.1}$ is used in this study to evaluate the model predictions
on an hourly and daily scale.

**2. Model description**
PMCAMx-UF is a Eulerian regional three-dimensional chemical transport model (Jung et al., 2010) that is an extension
of the PMCAMx model (Gaydos et al., 2007). The extended Dynamic Model for Aerosol Nucleation (DMANx) module
is used in PMCAMx-UF for the better description of ambient ultrafine particulate matter processes (Patoulias et al., 2015).
PMCAMx-UF solves the mass conservation equation for each pollutant in the gas, aqueous and particulate phases
focusing especially on the aerosol number and mass size distributions and the ultrafine particles.
Processes simulated by PMCAMx-UF include transport of pollutants via advection and eddy diffusion, their
chemical transformation in the gas, aerosol and aqueous (cloud) phases, their removal from the atmosphere through dry
(without water involvement) and wet (with water involvement) processes, their introduction into the atmosphere by direct
emission, whether from natural planetary processes or by human activity, and lastly specific physical processes for the
particle phase, namely coagulation, condensation/evaporation and nucleation. PMCAMx-UF simulates the temporal
variation of the complete aerosol number size distribution, beginning from particles as small as 0.8 nm and up to 10 μm
using 41 size bins. At the same time, the mass concentration of 18 major aerosol components is simulated, including
inorganics (ammonium, sulfate, metals, nitrate, sodium, chloride), primary and secondary organic aerosol, elemental
carbon and aerosol phase water. The secondary organic aerosol species are split into 4 volatility bins for the anthropogenic
and another 4 for those of biogenic origin. An extremely low volatility secondary organic aerosol (ELSOA) component
was added by Patoulias and Pandis (2022) to simulate the extremely low volatility secondary organic compounds.
Gas phase chemistry in PMCAMx-UF is described by the extended Statewide Air Pollution Research Center
(SAPRC) mechanism (ENVIRON, 2003; Patoulias and Pandis, 2022), which involves 219 thermochemical and
photochemical reactions, 64 gaseous compounds, of which 11 reactivity lumped organic compounds (5 alkanes, 2 olefins,
2 aromatics, a mono- and a sesqui-terpene) and 18 free radicals. PMCAMx-UF utilizes the variable sizes resolution
(VRSM) aqueous phase chemical module (Fahey and Pandis, 2001). The algorithm for horizontal advection is based on
the piecewise parabolic method of Colella and Woodward (1984) and its implementation by Odman and Ingram (1996).
Dry deposition is described by a first order kinetic removal rate. For gaseous pollutants, the dry deposition velocity is
calculated from the series resistance to impaction model of Wesely (1989). For aerosol species, the gravitational settling
velocity is in addition factored in. Its calculation follows the implementation of Slinn and Slinn (1980). Additional
information about PMCAMx-UF can be found in Patoulias et al. (2018).

Ultrafine particle levels, size distributions, and chemical compositions are shaped by the complex interplay of atmospheric processes such as nucleation, condensation of low-volatility compounds, condensation and evaporation of semivolatile compounds, coagulation, and direct emissions. Nucleation and condensation are critical for the formation and initial growth of new particles, whereas coagulation decreases particle number by removing smaller particles due to collisions with larger ones. Primary emissions, particularly from traffic and other combustion-related activities, are a major source of $PM_{0.1}$, especially in densely populated urban environments. Condensation is also a sink of $PM_{0.1}$ because it can lead to growth of nanoparticles to sizes larger than 100 nm. Xue et al. (2018) highlighted that combustion of natural gas and biogas can significantly contribute to atmospheric ultrafine particles. While CTMs can reasonably capture emissions and large-scale transport, considerable uncertainties persist in simulating nucleation processes, organic aerosol formation, and the removal mechanisms of ultrafine particles. Nucleation is expected to be minor to negligible source of $PM_{0.1}$ so the corresponding uncertainties in its simulation are expected to have a small effect on the accuracy of $PM_{0.1}$ predictions in continental areas. One of the objectives of this study is to obtain some insights into the ability of models like PMCAMx-UF to simulate the ensemble processes that drive $PM_{0.1}$ levels and variability.

**3. Model application**

PMCAMx-UF was applied to a modelling domain spanning the European continental area, covering a 5400x5832 $km^2$ area, using a rotated polar stereographic domain projection. This region is divided into 36x36 $km^2$ cells resulting in 24300 cells in each vertical level. In the vertical axis there are 14 levels, extending to approximately 7.2 km. The ground level, which is the main focus of this study, has a 60 m top boundary height.

The two periods examined correspond to 5 June to 8 July 2012 and 1 to 30 January 2009, during the PEGASOS and EUCAARI campaigns respectively. These periods have been selected because the corresponding emission inventories and meteorological inputs have been evaluated and improved in past modeling studies and the PMCAMx model has showed good performance in reproducing the $PM_{2.5}$ mass and composition (Skyllakou et al., 2014; Patoulias et al., 2018; Patoulias and Pandis, 2022. Inputs for this version of PMCAMx-UF for the two periods have been described by Patoulias and Pandis (2022).

Meteorological input data for both periods were generated by the Weather Research and Forecasting (WRFv2) model (Skamarock et al., 2005). This model utilizes geospatial time-varying meteorology data as inputs that are a product of the Global Forecast System (GFSv15) of the National Oceanic and Atmospheric Administration (NOAA). WRF model grids correspond to those of the chemical transport model. The original meteorological fields prepared by this older version of WRF have been evaluated in past studies and have been reused here to maintain consistency with these previous applications of PMCAMx and PMCAMx-UF. The more recent versions of WRF that offer improvements in model physics, computational efficiency, grid flexibility, and data assimilation capabilities will be used in future applications.

Anthropogenic particulate matter emissions have hourly space resolution and are based on the pan-European anthropogenic particle number emissions inventory and the carbonaceous aerosol inventory, both developed during the European Integrated project on Aerosol, Cloud, Climate, and Air Quality Interactions (EUCAARI) (Kulmala et al., 2011). These datasets include various anthropogenic sources such as ground transportation, shipping, industrial processes, domestic activities, etc. Anthropogenic gas-phase emissions are based on the Global and regional Earth-system Monitoring using satellite and in situ data (GEMS) inventory. Continental natural ecosystem emissions were derived using the Model of Emissions of Gases and Aerosol from Nature (MEGANv2.1) (Guenther et al., 2006). MEGAN requires

the meteorological inputs described above, as well as surface area type indicators. Natural marine emissions are based on the model of O'Dowd et al. (2008). Wildfire emissions included in our simulation were taken from the Sofiev et al. (2008a, b) emission inventory. Intermediate volatility organic compound emissions were estimated based on the primary organic aerosol emission rates, with proportionality factors depending on estimated volatility (Patoulias and Pandis, 2022) to maintain consistent inputs with previous studies. Murphy et al. (2023) have shown that it is better to estimate the IVOC emissions based on the total VOC emissions, instead of the POA. This approach will be used in future work.

Initial and boundary conditions used in this application were constant and low to minimize their influence on model predictions. The first two days of the summer and winter simulation periods are not included in the analysis. This is a time interval which has been shown to be adequate to exclude most of the influence of initial conditions in previous PMCAMx-UF applications (Patoulias et al., 2018; Patoulias and Pandis, 2022).

## 3.1 Measurements

Ultrafine particle mass is difficult to measure, primarily due to its low concentration. In order to evaluate hourly model predictions of ultrafine particulate matter concentrations, we use here surface level measurements of particle number size distributions, available through the EBAS database (https://ebas-data.nilu.no), during the Pan-European-Gas-AeroSol-climate interaction Study (PEGASOS) and the European Integrated project on Aerosol, Cloud, Climate, and Air Quality Interactions (EUCAARI) (Kulmala et al., 2011) intensive measurement campaigns. The locations of the 12 measurement sites are shown in Figure 1. These include Mace Head (Ireland), Varrio, Hyytiala (Finland), Aspvreten, Vavihill (Sweden), Helsinki (Finland), Waldhof, Melpitz, Dresden, Hohenpeissenberg (Germany), Kosetice (Czech Republic) and Finokalia (Greece). Particle number distribution measurements in each site were made through mobility particle sizers, either scanning (SMPS) or differential (DMPS). The ultrafine particle volume concentrations, $PV_{0.1}$, was then calculated by integrating these distributions up to 100 nm assuming spherical particles. We used this observed $PV_{0.1}$ directly for the model evaluation, because there were no available measurements of the chemical composition of the ultrafine particles was not available, and therefore it was not possible to estimate their density based on the measurements. In contrast, the model provides detailed information on the $PM_{0.1}$ composition, allowing us to calculate its predicted density. As a result, the $PV_{0.1}$ was the most appropriate variable for model evaluation in this study. For some sites, there were gaps in the available measurements. The corresponding analysis was based only on the days with available data for both measurements and predictions. As a result, these measurement gaps did not affect the model evaluation and corresponding conclusions.

The $PM_{0.1}$ predicted by PMCAMx-UF was converted to $PV_{0.1}$ by estimating the average ultrafine particle density, $\rho_{UFP}$, based on the predicted particle composition at each point at time:

$$PV_{0.1} = \frac{PM_{0.1}}{\rho_{UFP}}$$

$$(1)$$

$$\rho_{UFP} = \frac{\sum_{i=1}^{N} \rho_i \ PM_{0.1,i}}{PM_{0.1}}$$

$$(2)$$

where N is the total number of components, $\rho_i$ is the density of component $i$, $PM_{0.1,i}$ is the $PM_{0.1}$ mass concentration of component $i$, and the total $PM_{0.1}$ the total mass concentration.

Measurement uncertainties stem from both instrument limitations and the assumption that particles are spherical.
On the modeling side, inaccuracies primarily result from the predicted concentrations of $PM_{0.1}$ chemical composition and
the corresponding estimation of particle density. Additionally, the use of the 100 nm cutoff to define $PM_{0.1}$ introduces
some uncertainty, as this threshold is somewhat arbitrary. However, it was chosen to align with existing definitions and
to ensure consistency with previous studies. The U.S. Environmental Protection Agency (EPA, 2025) classifies ultrafine
particles as those smaller than 0.1 μm in diameter.

**4. Results**
**4.1 Average spatial variation of $PM_{0.1}$**
The average $PM_{0.1}$ predictions at the ground level during the summertime simulated period are shown in Figure 2. There
was considerable spatial variability of $PM_{0.1}$ levels throughout Europe. The mean value over the full domain (0.4 μg m$^{-3}$)
was heavily influenced by the fact that a significant part of the domain is over the Atlantic Ocean and Northern Africa,
regions with much lower concentrations of $PM_{0.1}$. Averaging without those parts and considering only the continental
regions of the domain, the average predicted $PM_{0.1}$ concentration was equal to 0.6 μg m$^{-3}$.
$PM_{0.1}$ was predicted to have higher values, up to 1.2 μg m$^{-3}$, in parts of southern and eastern Europe. High levels
were also predicted for major urban areas like Paris, as well as areas with high ship traffic like the North Sea or the
western Mediterranean. $PM_{0.1}$ was predicted to be, on average, 51% secondary inorganic matter (38% sulfate and 13%
ammonium), 41% organic matter (9% primary and 32% secondary), with smaller contributions from elemental carbon
(5%), metal oxides (2%) and trace contributions (<1%) of nitrate, sodium and chloride. Sulfate levels were higher in the
North Sea, the Mediterranean, parts of the Middle East and the Strait of Gibraltar, as well as the lower Bay of Biscay.
Ammonium spatial patterns mirror those of sulfate. SOA was a major $PM_{0.1}$ contributor in most of eastern and central
Europe. Primary organic aerosol (POA) and elemental carbon contributed relatively little mass on the domain scale, with
sharp spatial gradients in regions of increased human activity.
The average predicted $PM_{0.1}$ concentration and composition for the winter period are shown in Figure 3. The
average level over Europe was 0.3 μg m$^{-3}$ considering only continental regions and was lower than during the summer.
Wintertime $PM_{0.1}$ was predicted to consist of an average of 66% secondary inorganic material (47% sulphate and
19% ammonium), 23% primary matter (9% elemental carbon, 9% organic matter and 5% metals), with small amounts of
nitrate, sodium and chloride (<5%). SOA contributed 6% to the mean predicted $PM_{0.1}$, with higher contribution in
northwestern Russia, northern Italy and southern Spain and Portugal. The highest SOA average concentration was 0.1 μg
m$^{-3}$ in northwestern Russia. $PM_{0.1}$ in central and western Europe, as well as in key urban areas of the Iberian Peninsula
and northern Italy, was mainly composed of primary (emitted) matter. Primary matter concentration was as high as 0.9
μg m$^{-3}$ in urban areas. Sulfate, and the associated ammonium, were the major contributors to $PM_{0.1}$ in eastern Europe
according to PMCAMx-UF, however with reduced concentration relative to the summer. The $PM_{0.1}$ levels in northwestern
and central Europe were lower by around 0.2 μg m$^{-3}$ compared to the summer. In southern Italy, the concentrations were
reduced from more than 1 μg m$^{-3}$ to less than 0.4 μg m$^{-3}$. On the other hand, in many urban areas (e.g. Paris) the $PM_{0.1}$
levels were similar or even higher during the winter.

**4.2 Predicted $PM_{0.1}$ chemical composition in urban areas**
The average predicted chemical composition of $PM_{0.1}$ for selected sites is depicted in Figure 4 for the summer and winter
period. During the summer period, sulfate was a major $PM_{0.1}$ component, with its fractional mass contribution varying
from 17% to 52% depending on location, while SOA contributed from 18 to 50%. Ammonium (7-16%), primary organics
(4-18%), elemental carbon (2-30%) and metals (1-5%) were the remaining contributors. The mass percentage of sodium,
chloride and nitrate was in most sites less than 1%. The predicted $PM_{0.1}$ summertime concentration was mostly (52% to
91%) secondary (organic or inorganic). A significant fraction of the SOA (40-73%) was predicted to be anthropogenic in
all sites, 21-36% was predicted to be biogenic, and 7-25% was predicted to be extremely low volatility secondary organic
compounds (Table S3).
In summer, in the urban area of Athens, the major component of $PM_{0.1}$ was sulfate (33%), followed by SOA
(23%), primary organic aerosol (18%) and ammonium (13%). In Paris, elemental carbon had the highest contribution
(30%) to $PM_{0.1}$. Sulfate contributed 20% and SOA 20%. At the rural site of Finokalia, $PM_{0.1}$ consisted of 52% sulfate,
23% SOA and 17% ammonium, with smaller contributions of elemental carbon (2%) and primary organic aerosol (4%).
During the winter period, primary material contributed from 22% to 61% to $PM_{0.1}$ depending on location (Fig.
4). Primary organic aerosol ranged from 10% to 23%. Elemental carbon was predicted to contribute 8% to 31%, while
metals from 4% to 10% across all sites during this period. Ammonium and sulfate remained a significant fraction of $PM_{0.1}$
(33% to 69%), especially in the urban areas in eastern Europe. The sulfate fraction ranged from 24% to 49%, with
ammonium contributing from 9% to 20%. The contribution of SOA was limited, up to 9% at the sites examined. The
remaining $PM_{0.1}$ components, namely nitrate, chloride and sodium, were predicted to contribute up to 1% in almost all
the examined sites.
In Athens, wintertime $PM_{0.1}$ consisted of sulfate (37%), POA (23%), elemental carbon (15%) and ammonium
(13%). The remaining were metals (7%) and SOA (5%). In Paris, elemental carbon was the major $PM_{0.1}$ component with
a contribution of 30%, similar to summer, as transportation was its major source. Sulfate contributed 25%, while POA
20%. Lower contributions were predicted for ammonium (10%), metals (10%) and SOA (5%). In both Athens and Paris,
$PM_{0.1}$ was highly correlated with EC, especially during the periods with high $PM_{0.1}$ concentrations (Fig. S2). This was
also the case in other sites like Montseny, Zurich, Ispra, and Birmingham indicating the importance of combustion sources
for wintertime $PM_{0.1}$ and the significant contribution of elemental carbon made to $PM_{0.1}$ during the more polluted periods.
At the rural site of Finokalia, $PM_{0.1}$ mainly consisted of sulfate (49%) and ammonium (16%), with smaller contributions
of primary organic aerosol (10%), elemental carbon (8%), chloride and sodium.
During summer, the average chemical composition of $PM_{2.5}$ and $PM_{0.1}$ was similar in most areas as they were
both dominated by secondary components. SOA was the major component of $PM_{2.5}$ in most sites, contributing between
12% and 45%, with the highest levels in Zurich, Ispra, and Bucharest. Sulfate also played a significant role (13-34%),
particularly in Finokalia and Patras (Fig. S1). Ammonium contributed between 6% and 15% across all sites. Sulfate
contributed a little more to $PM_{0.1}$ than to $PM_{2.5}$ accounting for 30% to 50% of the $PM_{0.1}$ , while SOA and ammonium
contributions remained comparable to those in $PM_{2.5}$.
In winter, the composition of $PM_{2.5}$ was in general different from that of $PM_{0.1}$ in several cities, reflecting
differing major emission sources and formation mechanisms. POA contributed more to $PM_{2.5}$ (4-38%) than to $PM_{0.1}$ (10-
23%), whereas elemental carbon contributed less to $PM_{2.5}$ (2-17%) compared to $PM_{0.1}$ (8-31%) (Fig. S1). At coastal sites
like Patras, Finokalia, and Helsinki, secondary inorganic aerosol (including sulfate, nitrate, and ammonium) along with
crustal elements and sea salt, dominated the $PM_{2.5}$ composition, accounting for 82-90%. Sulfate concentrations were
generally lower $PM_{2.5}$ (17-34%) than in $PM_{0.1}$ fraction (24-49%) during winter.

**4.3 PMCAMx-UF evaluation**
*4.3.1 Summer*
During the summer period, PMCAMx-UF predictions showed on average little bias with a NMB equal to 15% for hourly
average concentrations (Table 1). The NME, on an hourly level, was on average 62%, a level similar to that of $PM_{2.5}$
predictions of CTMs in Europe. The model performance in this first application was clearly quite encouraging (Fig. S3).
NMB and NME hourly metrics in the various stations ranged from -29% to +109% and from +44% to +125%,
respectively. The model's performance improved, as expected, for daily average concentrations (Table S1). The NME
was reduced to 46%. The NMB remained at the low level of 15%.

279   During the summer, for most locations, model predictions as well as measured values exhibited significant

variability (Fig. 5). This spatial and temporal variability is mainly related to the spatial and temporal variability of
emission sources, secondary aerosol production and to the variability of meteorological conditions. In most sites, the
mean was larger than the median due to short-term elevated concentrations. PMCAMx-UF on average did a reasonable
job reproducing the observations, with overpredictions and underpredictions of $PV_{0.1}$, depending on the location. Average
concentrations for the full period were captured within 0.1 $\mu m^3$ $cm^{-3}$ for 7 out of 12 of the examined sites, with all the
predicted averages being within 0.25 $\mu m^3$ $cm^{-3}$ of measurements. Focusing on the urban sites, in Dresden, mean ultrafine
particle volume concentration was underpredicted by 0.17 $\mu m^3$ $cm^{-3}$. For Helsinki, the mean predicted $PV_{0.1}$ was quite
consistent with the measurements. In rural background areas (Vavihill, Aspvreten, Waldhof and Kosetice), PMCAMx-
UF overpredicted $PV_{0.1}$ by 0.13 to 0.25 $\mu m^3$ $cm^{-3}$. In general, predicted concentrations were higher than measurements.
Mean predicted $PV_{0.1}$ for all the sites examined was 0.34 $\mu m^3$ $cm^{-3}$ and the corresponding measured value was 0.29 $\mu m^3$
$cm^{-3}$.

291   In Dresden, the model predicted a weaker diurnal variation to that observed, but its main weakness was its

underprediction of the baseline by around 0.2 $\mu m^3$ $cm^{-3}$ (Fig. 6). A noticeable measured peak at 8:00 LST probably
indicates traffic emissions which were not captured in the model, either through omission or due to grid resolution. The
model tended overall to capture the hourly variations (Fig. S4), though it missed some high concentration periods on June
the 8, 10, 16 and 24.

296   For Helsinki, the average measured diurnal pattern was relatively flat (Fig. 6). Measured values were reproduced

well by PMCAMx-UF, with differences of around 0.05 $\mu m^3$ $cm^{-3}$ throughout most of the average day. The detailed time
series was also well reproduced (Fig. S4).

299   In Kosetice, for the first half of the day, predictions were far larger than the corresponding measurements, starting

the night at +0.1 $\mu m^3$ $cm^{-3}$ and peaking at 05:00-06:00 with a more than +0.2 $\mu m^3$ $cm^{-3}$ difference (Fig. 6). This increase
in predicted levels was due to an increase in traffic emissions. For the second half of the day, predicted and measured
values were in reasonable agreement. Excluding the first two days, which were influenced by the initial conditions, the
model overpredicted nighttime to early morning concentrations in several periods (June 10-12, 16-17, 24 and 26) (Fig.
S4). Measured concentrations were rarely higher than those predicted, for example on July 2 and 3, when sharp peaks
indicated possible nearby sources. The overprediction could indicate that emissions of UFPs in the area were
overestimated.

307 The average diurnal profiles of measured and predicted $PV_{0.1}$ concentrations as well as their corresponding

308 hourly levels for the rest of the 12 sites for the summer period can be found in Figure S4 and Figure S5. PMCAMx-UF

309 reproduced well the average diurnal profile of measured $PV_{0.1}$ in Hyytiala, with an average value of 0.25 $\mu m^3$ $cm^{-3}$, while

310 there were overpredictions during the whole day for Vavihill, Waldhof and Aspvreten.

311

312 *4.3.2 Winter*

313 PMCAMx-UF tended to underpredict the winter $PV_{0.1}$ levels with a NMB equal to -30% for hourly averaged values

314 (Table 2). The NME for hourly predictions was higher than during the summer with a value of 72%. For daily average

315 levels, the NMB was -27% and the NME equal to 64% (Table S2). The model overpredicted $PV_{0.1}$ by 0.03 to 0.09 $\mu m^3$

316 $cm^{-3}$ in the sites of Vavihill, Hyytiala, Aspvreten and Varrio.

317 Mean predicted values in 9 out of 12 sites were within 0.1 $\mu m^3$ $cm^{-3}$ of the measured mean (Fig. 7). $PV_{0.1}$ was

318 underpredicted in 7 out of 12 sites. Despite the increased frequency of underprediction, major positive deviations between

319 predictions and observations were found in the Varrio and Hyytiala sites, with high model error also in the Aspvreten,

320 Vavihill, Mace Head and Dresden sites. Mean predicted $PV_{0.1}$ was 0.17 $\mu m^3$ $cm^{-3}$ for all sites and mean measured $PV_{0.1}$

321 was 0.24 $\mu m^3$ $cm^{-3}$.

322 In Dresden, the ultrafine particle volume concentration was seriously underpredicted, 0.27 $\mu m^3$ $cm^{-3}$ to 1.22 $\mu m^3$

323 $cm^{-3}$ respectively. Mean ultrafine particle volume concentration for Helsinki was also underpredicted, with a predicted

324 value of 0.18 $\mu m^3$ $cm^{-3}$ and a measured value of 0.35 $\mu m^3$ $cm^{-3}$. On the other hand, for the remote Hyytiala site in Finland,

325 mean predicted total $PV_{0.1}$ was 0.16 $\mu m^3$ $cm^{-3}$, compared to a measured average of 0.07 $\mu m^3$ $cm^{-3}$. This suggests that the

326 underpredictions in Helsinki were mostly due to local sources and not to regional underprediction.

327 In Dresden, the measured levels increased by a factor of two early in the morning while the predicted profile

328 remained practically flat (Fig. 8). This suggests strongly the lack of one or more major local sources, probably

329 transportation and residential heating. It could also be partially due to the coarse resolution of the model; local emissions

330 were diluted in the large computational cell of the model covering the area of the city. The corresponding hourly

331 concentrations are shown in Figure S6.

332 For Helsinki, the predicted average diurnal profile was nearly flat (variation less than 0.05 $\mu m^3$ $cm^{-3}$) throughout

333 the day, while the measurements peaked at 10:00, remaining near constant during midday and then gradually decreasing

334 (Fig. 8). The hourly concentrations suggested that the model was rarely able to reproduce observed elevated concentration

335 levels during specific one to two-day periods (Fig. S6). The sources of ultrafine particles during these periods need to be

336 further examined. Errors in the meteorological inputs and especially the mixing height were also a possible explanation

337 of these persistent errors.

338 In Hyytiala, the diurnal average profiles of measured and predicted values were both flat but they differed by

339 approximately 0.1 $\mu m^3$ $cm^{-3}$ (Fig. 8). This suggests that the model agreed with observations regarding the relatively low

340 local contributions but it overpredicted the regional background. This could be partially due to the assumed boundary

341 conditions that influenced the Nordic countries more than the rest of Europe due to the choice of modeling domain.

342 Turning our attention to the full period hourly concentrations, substantial deviations became readily apparent (Fig. S7).

343 For the first half of the simulated period, predicted UFP volume concentrations tended to follow measured values, with

344 rapid increases in measured concentrations not generally predicted. These were again possibly indicative of local sources

345 influencing the measurement site. After January 17, the model overpredicted $PV_{0.1}$. The reasons for this overprediction

require future analysis. The corresponding hourly $PV_{0.1}$ concentrations as well as their average diurnal profiles for the rest
of the 12 sites for this winter period can be found in Figure S6 and Figure S7.
Average volume distributions for measured and predicted $PV_{0.1}$ were in general consistent with a monotonically
increasing shape (Figure S8). For sites in which PMCAMx-UF was in good agreement with the $PV_{0.1}$, the measured size
distributions were also in good agreement for all sizes, suggesting that the good performance of the model was not due to
offsetting errors. In most areas where there were discrepancies the predicted size distribution was correct but there were
errors in the magnitude. Dresden during the winter was the exception, with the measured volume distribution starting to
increase at 15 nm while the predicted one started to rise at 30 nm. This suggests that the model was missing a major
ultrafine particle source in this site during the cold period. In all sites the predicted and measured volume distributions
suggested that nucleation made a minor contribution to ultrafine particle mass concentrations.
The spatial and seasonal variation in $PM_{0.1}$ concentrations is largely driven by emission patterns, which fluctuate
across different timescales -from monthly to hourly. The geographic distribution of these emissions, influenced by land-
use characteristics across the study area, contributes to regional differences. Weather conditions also have a strong
influence, with variables like wind speed and direction, boundary layer height, and solar radiation affecting how particles
are dispersed, transported, formed and removed. Additionally, photochemical processes are a key factor, as a substantial
portion of $PM_{0.1}$ is produced in the atmosphere from gas-to-particle conversion processes, making chemical reactivity and
sunlight-driven transformations major contributors to its variability.
The depth of our analysis of the evaluation of PMCAMx-UF for $PM_{0.1}$ is at present limited by the lack of
measurements of the chemical composition of $PM_{0.1}$ and the related measurement-based source apportionment studies in
Europe. This limits our ability to reach firm conclusions about what the model gets right and where it fails. For a lot of
the aspects of $PM_{0.1}$ behavior (e.g., composition and sources) our work presents our present understanding based on model
predictions (emissions and atmospheric processes) to motivate and help in the design of future studies.

**4.4 Predicted links between $PM_{0.1}$ and $PM_{2.5}$**
Current regulations are focusing on the reduction of $PM_{2.5}$. It is not clear if these strategies will be effective in the reduction
of $PM_{0.1}$ too. One way to address this issue at least as a first step is to examine the temporal correlation between $PM_{0.1}$
and $PM_{2.5}$. A correlation would suggest that the sources and processes driving particle mass concentrations in both size
ranges are similar, and therefore control strategies that will work for $PM_{2.5}$ will also be effective for $PM_{0.1}$. Low
correlations would suggest that different approaches may be needed for the reduction of both fine and ultrafine particle
mass.
The correlation of predicted $PM_{2.5}$ with $PM_{0.1}$ was examined during the summer and winter period. For the
summer period, the mass concentration of fine and ultrafine particles had low correlation in Zurich, Bucharest and
Helsinki, with comparatively better correlations in Athens, Birmingham and Paris (Fig. 9). In Helsinki, the two values
have a coefficient of determination ($R^2$) of 0.01. Ultrafine particle mass in Helsinki, as well as in Bucharest and Zurich
was mostly secondary inorganic and organic during the summer period. In Athens, Paris and Birmingham, the correlation
was significantly better, around 0.4 to 0.6. For Athens, the correlation was driven by wildfire episode (Fig. S9). If this
period is excluded the correlation decreases significantly.
For the winter period, correlations were high across most major cities examined, with the notable exceptions of
Bucharest and Birmingham (Fig. S10). The $R^2$ for Zurich, Birmingham, Bucharest and Helsinki was less than or equal to
0.4, but it was higher for Athens (0.71) and Paris (0.65).
For most major cities, an increase in the primary component of $PM_{0.1}$, was accompanied by an increase in its
correlation with $PM_{2.5}$. The exceptions were again Birmingham and Bucharest. The predicted $R^2$ value in both cities seems
to be influenced by outliers of substantially elevated $PM_{2.5}$ values. Yu et al. (2019) reported an $R^2$ between predicted $PM_{2.5}$
and $PM_{0.1}$ in a year-long study in California, for all domain cells, of 0.63. In that study, $PM_{0.1}$ was mostly comprised of
primary matter from combustion processes. This value is comparable to the highest observed in our study, specifically in
Athens and Paris.
The correlation between $PM_{0.1}$ and $PM_{2.5}$ was typically weak, but stronger associations were found when the
primary component of $PM_{0.1}$ played a significant role. This suggests notable differences in the sources and processes that
contribute to $PM_{0.1}$ and $PM_{2.5}$.

**5. Conclusions**
Predicted levels of $PM_{0.1}$ were quite variable in space and time. The average predicted total $PM_{0.1}$ for the continental
regions over Europe was 0.6 µg m$^{-3}$ for the summer and 0.3 µg m$^{-3}$ for the winter period. On average, sulfate (38%), SOA
(32%), ammonium (13%) and POA (8%) were the most significant $PM_{0.1}$ components during the summer. Primary and
secondary inorganic matter had an increased mass fraction (16% to 23% and 51% to 66%) during the winter period. The
secondary organic matter percentage contribution was quite low (6%) during the winter. The high secondary contribution
to $PM_{0.1}$ is rather surprising.
$PM_{0.1}$ during the winter period correlates better ($R^2$=0.18-0.71) with $PM_{2.5}$ than during the summer period
($R^2$=0.01-0.6). However, for most major cities the correlation is low. Better correlations were observed in cases where
primary sources contributed significantly to $PM_{0.1}$.
PMCAMx-UF showed little bias (15%) in reproducing the summertime ultrafine volume observations in 12 sites
in Europe. During the winter, the model tended to underpredict $PM_{0.1}$ with a NMB of -30% for hourly average values.
The model NME for daily average levels was 46% during the summer and 64% during the winter. Using the CTM
performance criteria for $PM_{2.5}$, the model performance was considered good for the summer and average for the winter.
Missing winter sources and processes need additional investigation.
Given that this is the first effort to predict $PM_{0.1}$ in Europe with PMCAMx-UF, the model performance was quite
encouraging. Potential model improvements include corrections in emissions especially during the winter, use of higher
grid resolution for the major urban areas and revisiting of the boundary conditions over the northern Atlantic. Evaluation
of its composition predictions is also needed. Future work will focus on more recent periods, providing a more detailed
analysis of not only total $PM_{0.1}$ concentration but also the contribution of individual sources.
The predicted lack of correlation between ultrafine and fine particle mass concentration suggests different
sources and processes and that future emission reduction strategies will have different effects on $PM_{0.1}$ and $PM_{2.5}$. For
example, sources which tend to emit smaller particles will have a larger impact on $PM_{0.1}$ than $PM_{2.5}$. Condensation of
secondary material will increase $PM_{2.5}$ but it may decrease $PM_{0.1}$ by growing particles outside the ultrafine particle range.
Coagulation is also expected to be a net sink for $PM_{0.1}$ as the small particles in this size range collide with larger particles
mainly in the accumulation mode. Coagulation has a minor effect on $PM_{2.5}$ because under most conditions it does not

transfer mass outside this size range. The analysis of the processes and sources that affect $PM_{0.1}$ will be examined in detail in future work. The main objective of the present work has been to lay the foundation for such a study by demonstrating that we can simulate $PM_{0.1}$ with a reasonable level of accuracy and therefore it makes sense to use the corresponding CTM for more detailed process analysis and source attribution.

*Code and Data Availability.* The model code and data used in this study are available from the authors upon request (spyros@chemeng.upatras.gr).

*Author Contributions.* KM carried out the simulations, the analysis, ES wrote the final manuscript with support from SNP., KM and DP, SNP supervised and coordinated the work.

*Competing Interests.* The authors declare no competing financial interest.

*Acknowledgements.* This work was supported by «Atmospheric nanoparticles, air quality and human health», NANOSOMs (11504) and the EU H2020 RI-URBANS (grant 101036245) project.

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

**Table 1.** PMCAMx-UF hourly evaluation metrics of $PV_{0.1}$ during the period of 5 June - 8 July 2012 for the 12
measurement sites.

| Station | Mean Predicted ($\mu m^3$ $cm^{-3}$) | Mean Observed ($\mu m^3$ $cm^3$) | NMB (%) | NME (%) |
|---------|---------------|--------------|---------|---------|
| Dresden | 0.42 | 0.59 | -29 | 45 |
| Kosetice | 0.37 | 0.24 | 54 | 82 |
| Hohenpeissenberg | 0.22 | 0.27 | -19 | 49 |
| Mace Head | 0.05 | 0.06 | -5 | 81 |
| Finokalia | 0.39 | 0.36 | 6 | 47 |
| Vavihill | 0.47 | 0.28 | 66 | 82 |
| Helsinki | 0.44 | 0.48 | -9 | 44 |
| Melpitz | 0.41 | 0.33 | 21 | 61 |
| Hyytiala | 0.22 | 0.23 | -3 | 61 |
| Waldhof | 0.50 | 0.31 | 63 | 81 |
| Aspvreten | 0.48 | 0.23 | 109 | 125 |
| Varrio | 0.10 | 0.10 | -8 | 68 |

















**Table 2.** PMCAMx-UF hourly evaluation metrics of $PV_{0.1}$ during the period of 1-30 January 2009 for the 12 measurement sites.

| Station | Mean Predicted ($\mu m^3 \, cm^{-3}$) | Mean Observed ($\mu m^3 \, cm^{-3}$) | NMB (%) | NME (%) |
|---|---|---|---|---|
| Dresden | 0.27 | 1.22 | -78 | 78 |
| Kosetice | 0.24 | 0.46 | -47 | 56 |
| Hohenpeissenberg | 0.16 | 0.18 | -16 | 51 |
| Mace Head | 0.02 | 0.11 | -78 | 82 |
| Finokalia | 0.07 | 0.14 | -48 | 65 |
| Vavihill | 0.25 | 0.20 | 27 | 83 |
| Helsinki | 0.18 | 0.35 | -50 | 66 |
| Melpitz | 0.27 | 0.28 | -6 | 52 |
| Hyytiala | 0.16 | 0.07 | 130 | 187 |
| Waldhof | 0.27 | 0.27 | 3 | 53 |
| Aspvreten | 0.11 | 0.08 | 33.5 | 114 |
| Varrio | 0.09 | 0.02 | 399 | 436 |


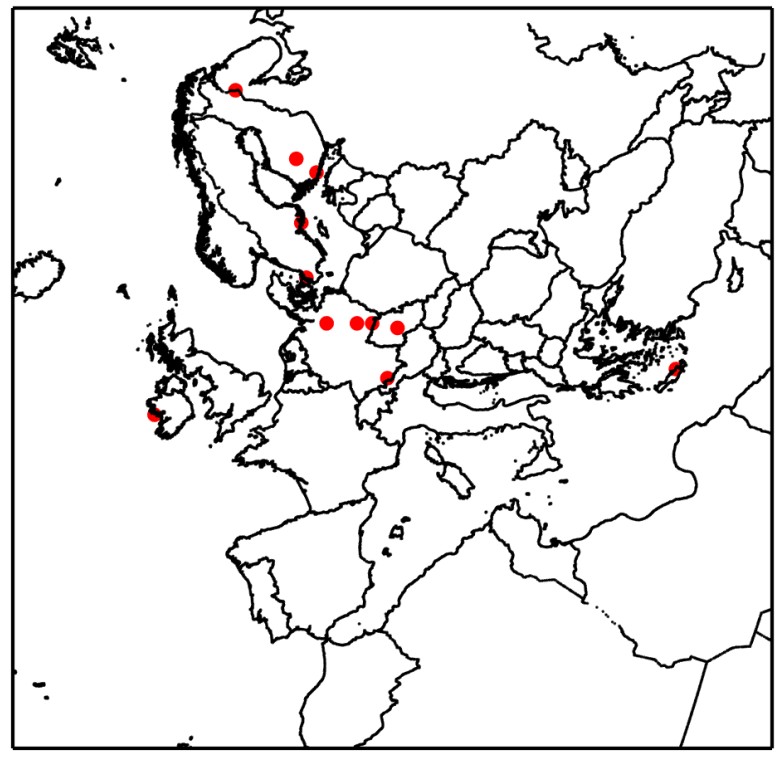

**Figure 1.** Map of the European modelling domain indicating (red dots) the 12 measurement sites with available particle
number distribution measurements for both simulation periods.


















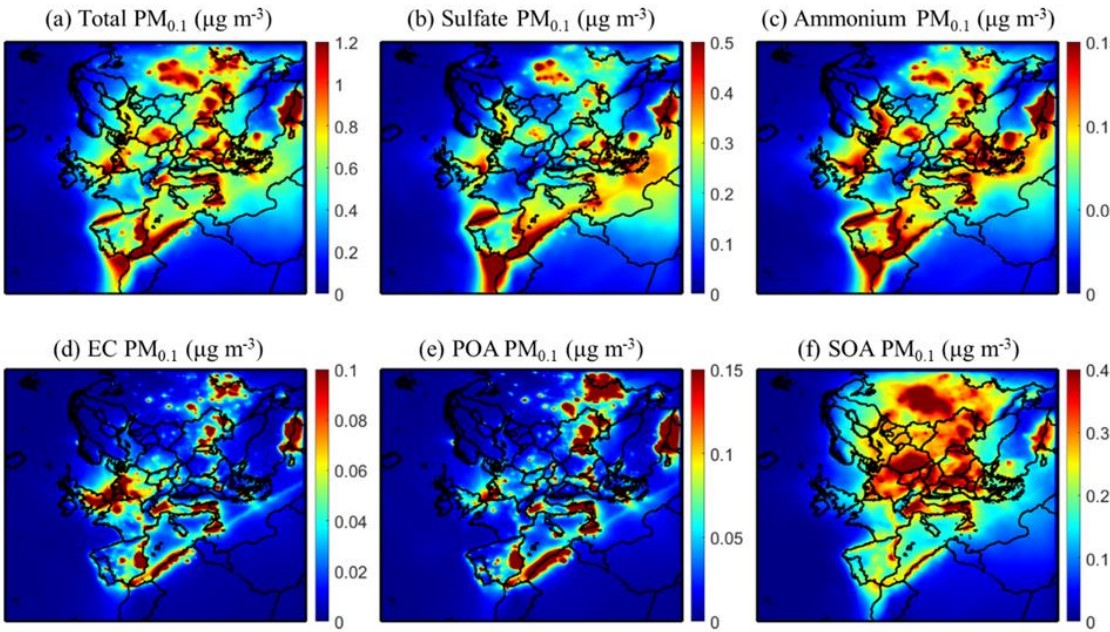


**Figure 2.** Average predicted ground level PM$_{0.1}$ mass concentrations (µg m$^{-3}$) of (a) total PM$_{0.1}$, (b) PM$_{0.1}$ sulfate, (c)
PM$_{0.1}$ ammonium, (d) PM$_{0.1}$ elemental carbon, (e) PM$_{0.1}$ primary organic aerosol and (f) PM$_{0.1}$ secondary organic aerosol
during 5 June - 8 July 2012.

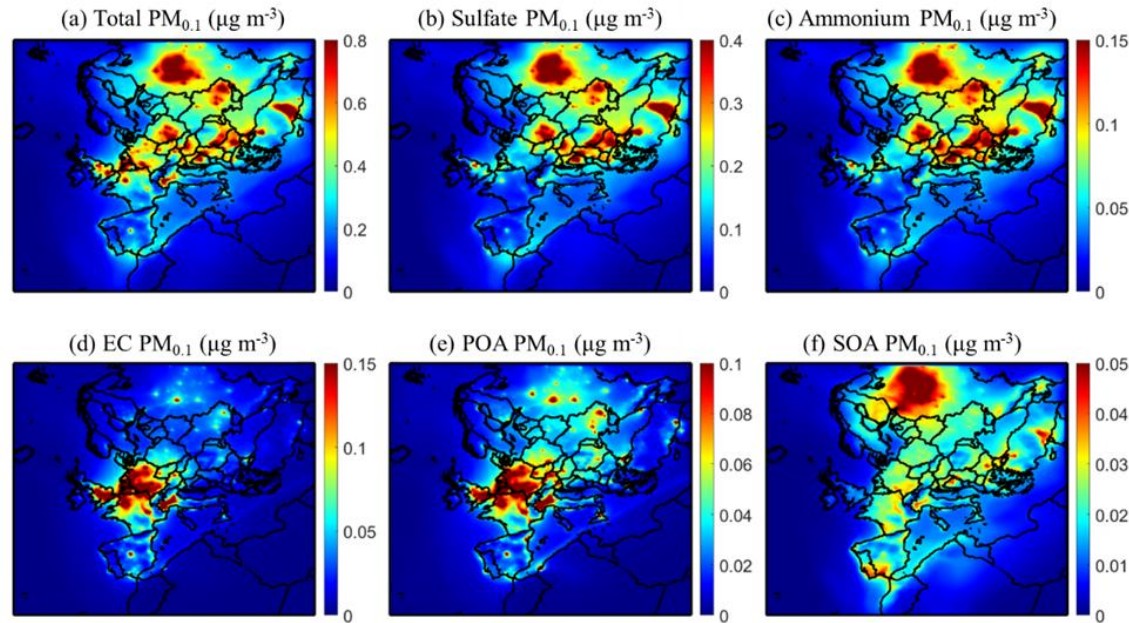

**Figure 3.** Average predicted ground level PM$_{0.1}$ mass concentrations (µg m$^{-3}$) of (a) total PM$_{0.1}$, (b) PM$_{0.1}$ sulfate, (c) PM$_{0.1}$ ammonium, (d) PM$_{0.1}$ elemental carbon, (e) PM$_{0.1}$ primary organic aerosol and (f) PM$_{0.1}$ secondary organic aerosol during 1 - 30 January 2009.

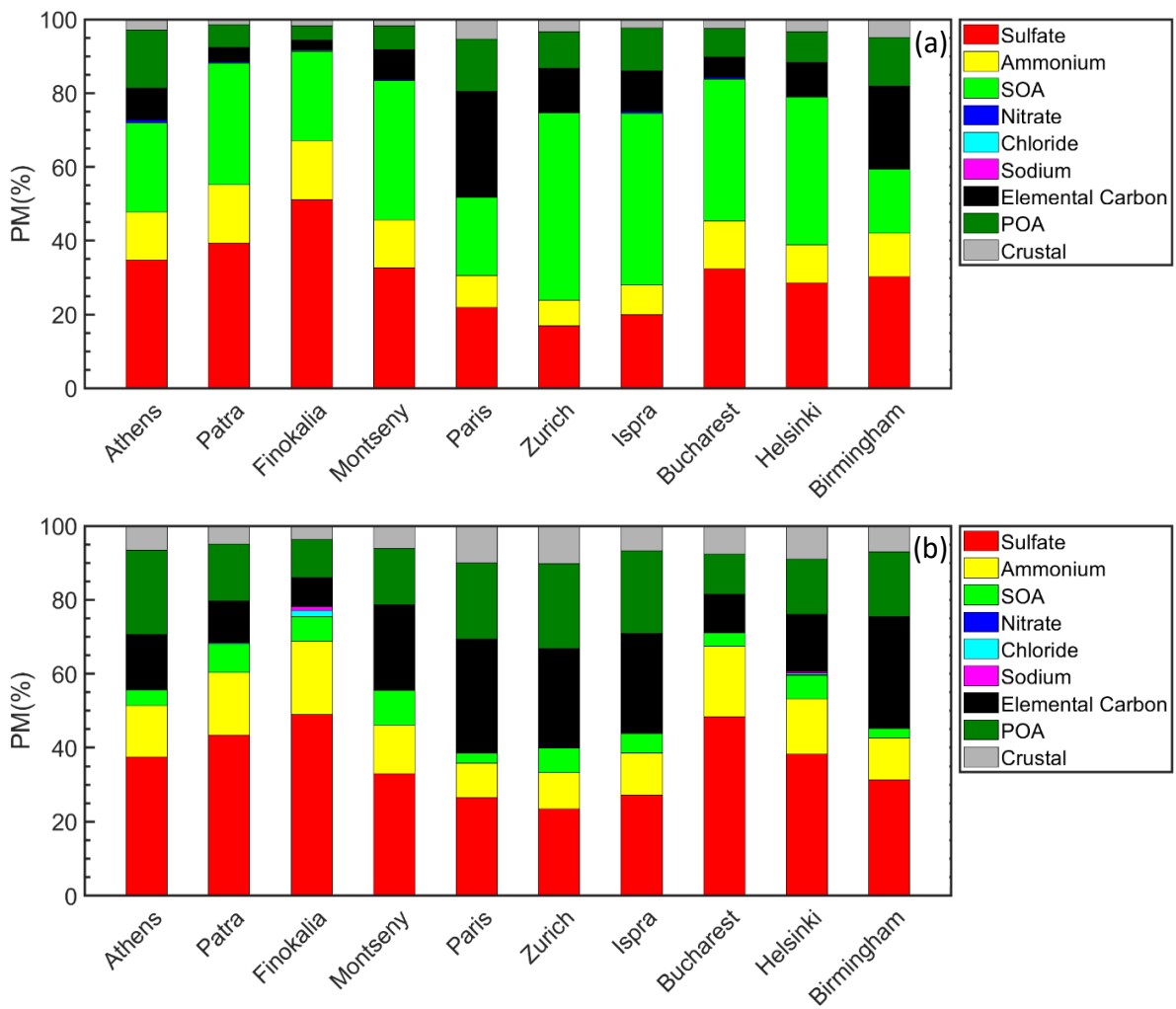

681

**Figure 4.** Predicted chemical composition of ultrafine particles in the areas studied during the (a) summer and (b) winter period. POA (dark green) and SOA (green) stand for primary and secondary organic aerosol.

684

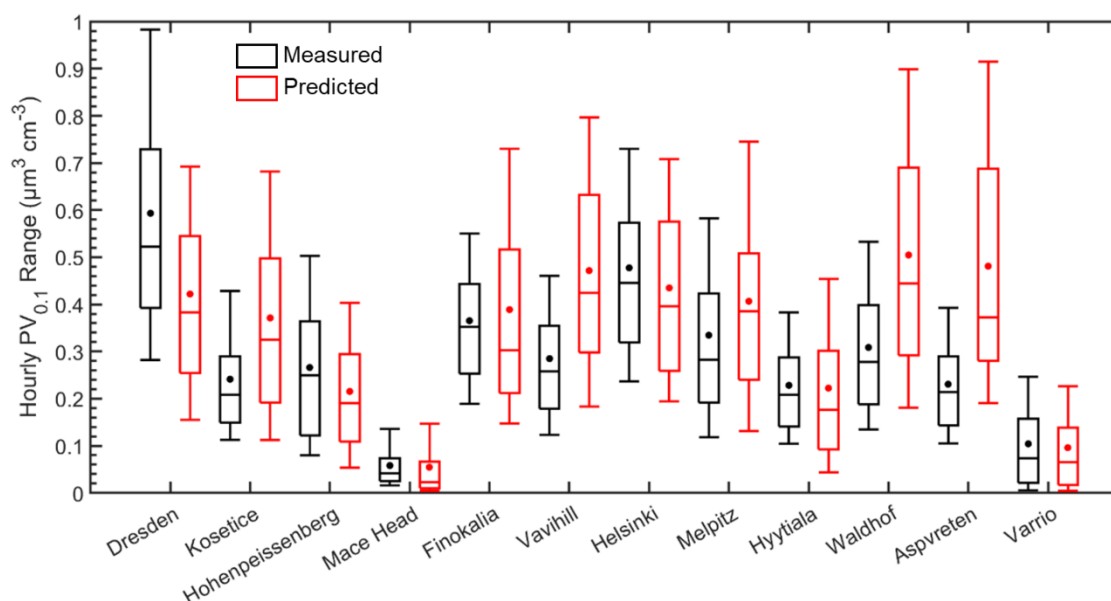

**Figure 5.** Distributions of predicted (red) and measured (black) hourly ground-level UFP volume (in $\mu m^3\ cm^{-3}$) during 5 June - 8 July 2012, in the 12 sites examined. Stars and lines inside each box designate the mean and the median value of the $PV_{0.1}$ distribution. Box top and bottom lines indicate the upper (75%) and lower (25%) quartiles. The upper and lower extended lines (whiskers) are for the 90th and the 10th UFP volume distribution percentiles.

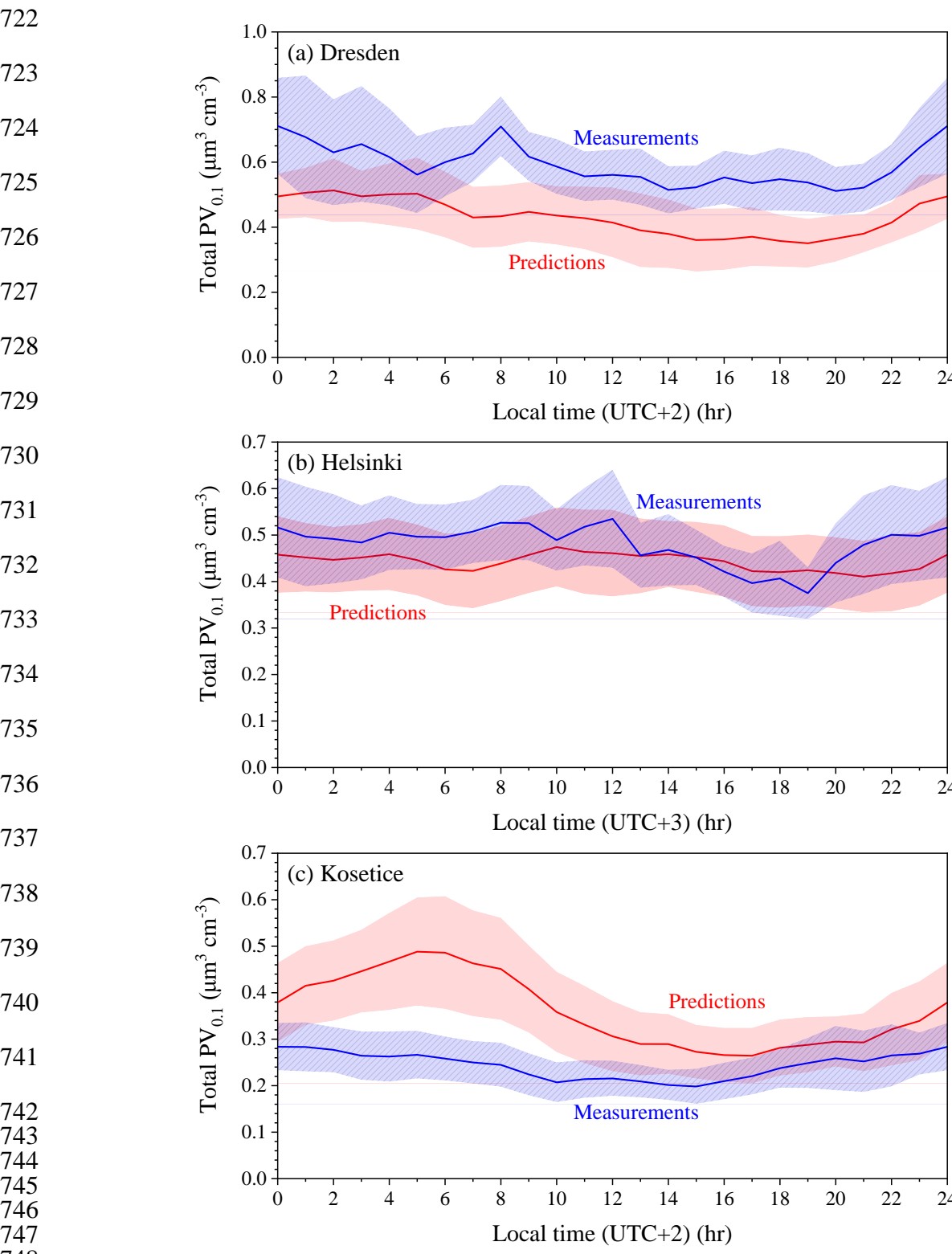

**Figure 6.** Average diurnal profiles of predicted and measured total volume concentrations ($\mu m^3$ $cm^{-3}$) in (a) Dresden, (b) Helsinki and (c) Kosetice for the period of 5 June - 8 July 2012. The shaded regions reflect plus or minus one standard deviation of the mean.

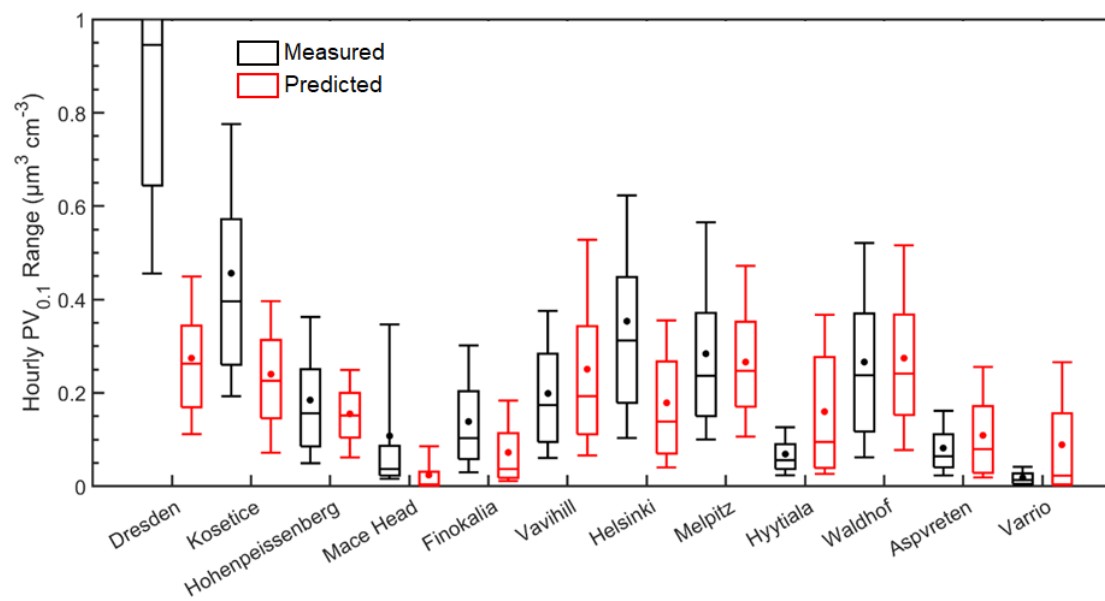

**Figure 7.** Distributions of predicted (red) and measured (black) ground-level UFP volume during 1-30 January 2009, in the 12 sites examined. Stars and lines inside each box designate the mean and the median value of the $PV_{0.1}$ distribution. Box top and bottom lines indicate the upper (75%) and lower (25%) quartiles. The upper and lower extended lines (whiskers) are for the 90th and the 10th UFP volume distribution percentiles.

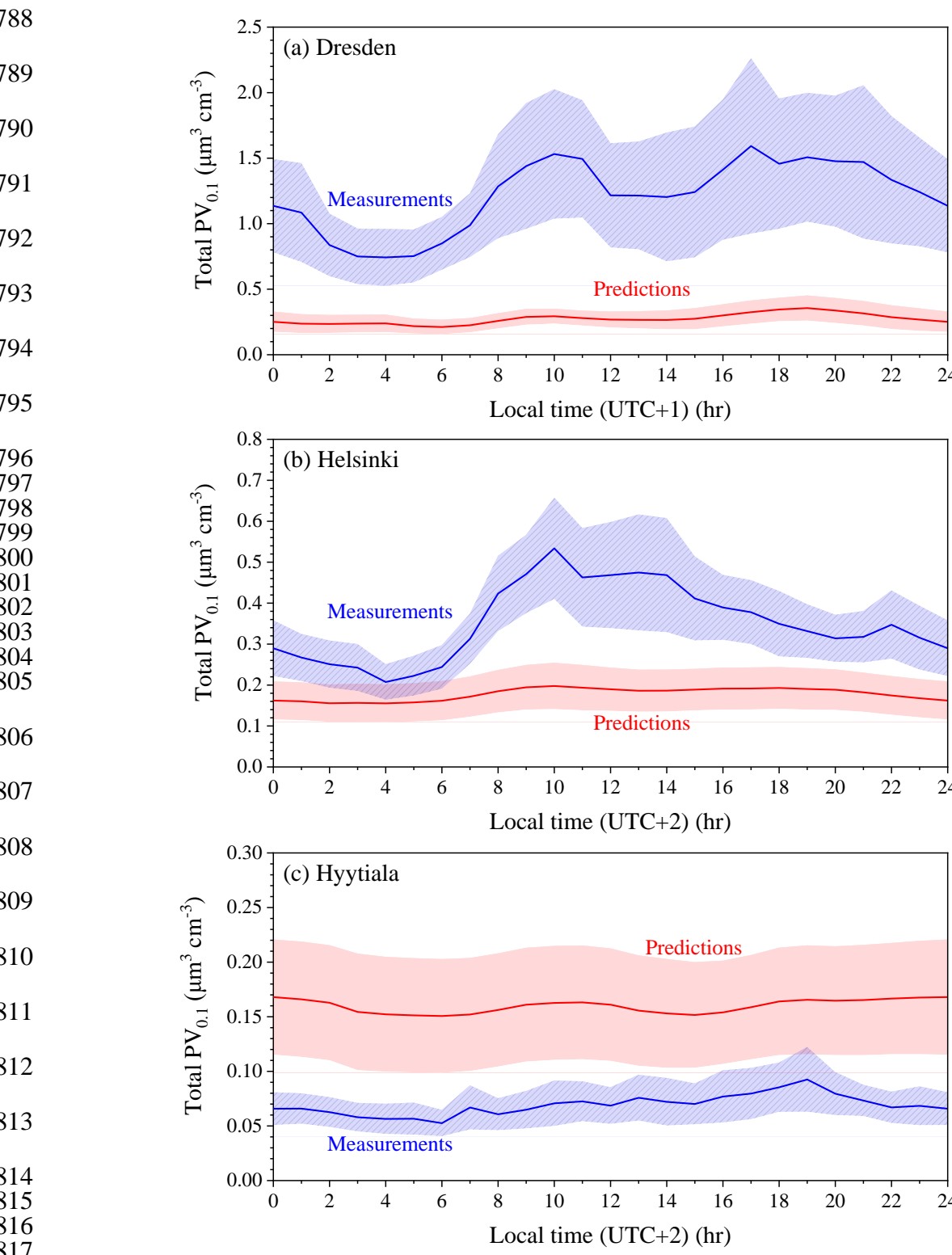

**Figure 8.** Average diurnal profiles of predicted and measured total volume concentrations ($\mu m^3$ $cm^{-3}$) in (a) Dresden, (b) Helsinki and (c) Hyytiala for the period of 1-30 January 2009. The shaded regions reflect plus or minus one standard deviation of the mean.

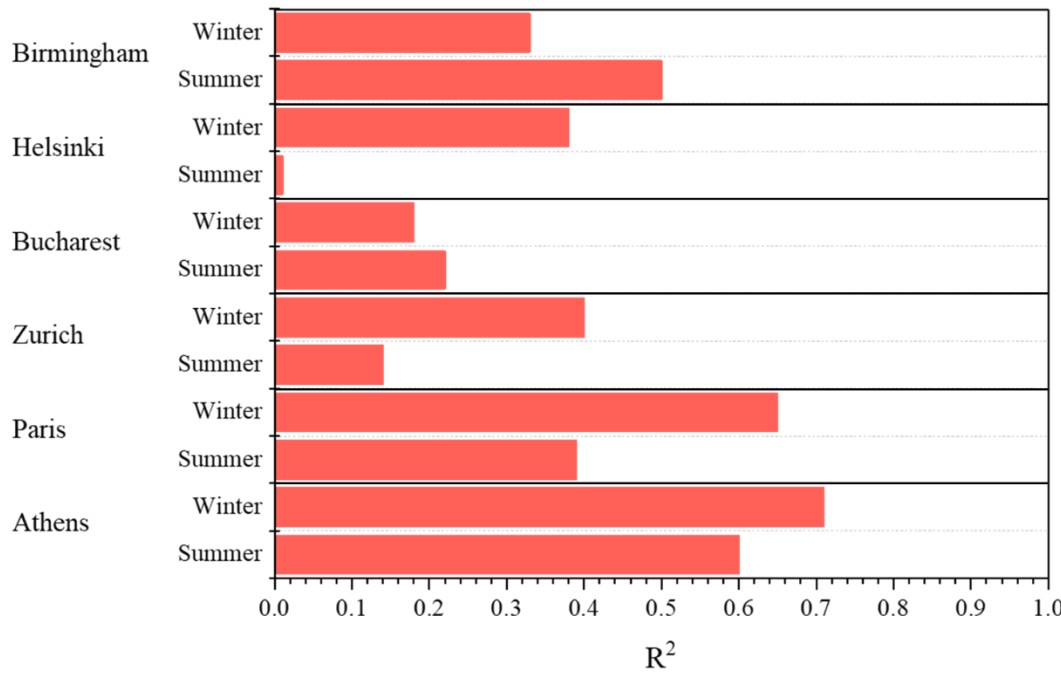

**Figure 9.** $R^2$ values (square of the samples Pearson's correlation coefficient) between $PM_{0.1}$ and $PM_{2.5}$ for Athens, Paris,
Zurich, Bucharest, Helsinki and Birmingham during the summer and winter periods.
