# Peer review of "Simulation of the seasonal and spatial variability of the 2 concentrations and chemical composition of ultrafine particulate 3 matter over Europe"

_EGUsphere, 2024_

## Author Response (AR1)

**Responses to Comments of Reviewer 1**

*General comment*

**(1)** Mataras et al. present in their manuscript a study on the characterization of ultrafine particulate matter ($PM_{0.1}$) mass concentrations and their chemical composition during a summer and winter period in Europe. Ultrafine particles have attracted interest because they may be the most dangerous fraction of atmospheric particulate matter. In general, very few modelling attempts aimed at ultrafine particle mass have been conducted, especially over Europe. The Authors used PMCAMx-UF model, a Eulerian regional three-dimensional chemical transport model that is an extension of the PMCAMx model. They predicted/modelled the levels of $PM_{0.1}$ and its chemical composition over Europe and specifically at a series of 12 urban and rural sites. They, further, evaluated the performances of the PMCAMx-UF model for daily average ultrafine particle volume ($PV_{0.1}$) concentrations. I think the paper is well written and it is neatly exposed. The literature cited is adequate and so are the graphics. It presents valuable results and conclusions and contributes to the growing knowledge on ultrafine particles pollution. Therefore, I consider this paper suitable for publication on Atmospheric Chemistry and Physics. Given the paucity of published model works about ultrafine mass concentration and especially in Europe region, this paper can represent a useful addition to the field if these issues are addressed.

I suggest a minor revision of the manuscript according to my specific comments before to consider it for publication in the Journal. There are a few issues that may warrant some additional thought and perhaps some further analysis.

We appreciate the positive assessment of our manuscript, as well as the constructive comments and suggestions. Below, we address the reviewer's major concerns. Our responses and the corresponding changes in the manuscript (in black) follow each of the reviewer's comments (in blue).

*Specific comments*

**(2)** **Title**: The title of the manuscript does not seem to fully reflect the core content of the study. It lacks key terms that would help identify the specific focus and contributions of the work. I suggest revising the title to better align with the main findings and scope of the research.

We have improved the title for our manuscript to reflect better the content of the study.

**(3)** In Line 101-102 the authors stated that an extremely low volatility secondary organic aerosol (ELSOA) component was simulated and then related results were reported in Table S3. However, nothing was reported and discussed about it in the manuscript. Why? It should be important to discuss this component.

We have added information about the contribution of ELSOA in the revised manuscript.

**(4)** From Figures S2 and S4, it is evident that several time series (i.e. Hohenpeissenberg, Aspvreten, Dresden, Finokalia) have a temporal gap and they are incomplete for the period under consideration. This could certainly affect the comparison between the measurements and the model outputs. It might be worth addressing this issue in your work or at least explicitly mentioning it in the article.

Some time series have temporal gaps due to lack of measurements. We now clarify that the comparisons between measurements and predictions and the corresponding

evaluation metrics for each of these sites refer to exactly to the same periods for which observations are available. As a result, the measurement gaps do not affect the evaluation and the corresponding conclusions.

**(5)** Throughout the manuscript, several acronyms are not explicitly defined, which may affect readability. I recommend defining each acronym upon first use to improve clarity for the reader.

In the revised manuscript, all acronyms are defined when they are first used.

**(6) Abstract**: Line 12-14. This sentence appears to mislead the reader regarding the true focus of the study. The work is not centered on measuring the mass concentration of ultrafine particles but rather on modelling these concentrations and evaluating the model's performance through a comparison with measurements provided in literature. I recommend revising this sentence to better reflect the core objectives and contributions of the study.

We have rewritten this sentence to avoid misunderstandings.

**(7)** Line 65-66. Authors reported as a reference Bernardoni et al, 2017, but their measurements were in north Italy and not in California.

We have removed the relevant reference from this point of the text.

**(8)** Line 73-75: This sentence may be misleading for the reader. Studies on ultrafine particles outside the United States are numerous and cover a wide range of topics beyond just number concentration or size distribution. Perhaps the authors intended to refer specifically to the mass concentration of ultrafine particles.

We have rewritten this sentence focusing on the mass concentration and composition of $PM_{0.1}$.

**(9)** Line 151-156. The authors use the volume concentration ($PV_{0.1}$) directly to avoid estimating the average density of ultrafine particles when converting to mass concentration. However, they later take the mass concentration from the model output, estimate the density based on chemical composition, and convert it back to volume concentration. This approach raises questions about its advantage, as the added steps seem to undermine the initial reasoning for using $PV_{0.1}$ directly. It seems that this approach is taken because there is no measured data on chemical composition to derive the particle density, while the predicted chemical composition from the model is available. However, this reasoning should be explicitly stated, as the phrase 'to avoid complications' is too vague and does not clearly justify the methodological choice.

We have followed the suggestion of the reviewer, and we now clarify in the revised manuscript that we used the measured volume concentration directly due to the lack of information on the chemical composition of measured ultrafine particles, which prevented the estimation of their density. In contrast, for the model, the exact chemical composition of ultrafine particles is available, so the density of the predicted $PM_{0.1}$ can be estimated.

**(10)** Line 204, 214. Here (line 204), it is stated that elemental carbon in Paris is the dominant component, accounting for 30%, and the same behavior (line 214) is observed in winter. Is this just a coincidence, or is there an underlying explanation for this pattern?

According to the predictions of PMCAMx-UF, the high predicted levels of elemental carbon are due to the high transportation emissions in the inventory used for the megacity in both seasons. This explanation has been added to the paper.

**(11)** Line 235-237. This sentence is ambiguous and may confuse the Reader. It is unclear what it refers to and seems to contradict the statement in line 232. I suggest clarifying it to ensure consistency and avoid misunderstanding.
We have deleted the confusing sentence and revised the others in this paragraph.

**(12)** Line 253-254. In this sentence it should be convenient to explicit that the UFPs emission were "overestimated" and not "overpredicted" to avoid confusion.
We have followed the suggestion of the reviewer to improve the accuracy of this statement.

**(13)** Paragraph 4.4. The objective of the study presented in this paragraph is not entirely clear. The authors provide a detailed discussion of the comparison results between $PM_{0.1}$ and $PM_{2.5}$ (ultrafine and fine particles), but the ultimate purpose of this comparison remains rather unclear. It might be helpful to better explain the aim of this analysis in the text. While the purpose of this comparison is somewhat explained in the Conclusions (lines 334-335), it would be beneficial to elaborate on it more clearly in this paragraph to provide better context and understanding for the Reader.
In the revised manuscript, the objective of this comparison is clearly stated in the beginning of the section to provide the necessary context for the reader.

**(14)** Figure 4. The labels (a) and (b) are missing between the two panels.
We have added the corresponding labels to the revised figure.

**(15)** Figure 6. In this figure it should be convenient for the Reader to insert the standard error (as error bars) in the measured data and an error bar for the modelled data.
We have revised Figure 6 by adding standard errors for both measured and predicted values.

**(16)** Figure 8. As in Figure 6.
We have added the corresponding error values in Figure 8 too.

**(17)** Figure 9. In the caption it should be convenient to explicit that $R^2$ is the Pearson coefficient of the scatter plots reported in Figure S5.
We have added this explanation in the corresponding figure caption.

**(18)** Figure S1. As in Figure 6.
We have also revised this figure adding the error ranges.

**(19)** Figure S3. As in Figure 6.
Figure S3 has been revised.

**Responses to Comments of Reviewer 2**

*General comments*

**(1)** Mataras et al. have performed simulations with a chemical transport model (PMCAMx-UF) to compare mass/volume concentrations and composition for particles smaller than 0.1 µm at several cities across Europe over 2 distinct seasons. They find the model performance is modest in some cities with specific biases in other cities over winter. The authors provide strong motivation for this work, including a robust set of methods to answer the questions posed. A minor objection to the study is the use of dated episodes and model versions, both of which could have been updated to present findings for the modern atmosphere. My primary concern, however, is that the paper runs a single configuration of the model and compares predictions to evaluations on mostly a time-averaged basis (over the episode) across a dozen sites. The text reads more like a technical report with little depth in analysis and interpretation. In my view, this is a missed opportunity and a lot more insight could have been extracted from these comparisons. With some effort, it should highlight what the model gets right and where it fails, with implications for how emissions, chemistry, and deposition are represented in chemical transport models, especially for particles smaller than 0.1 µm. At this point, I do not recommend publication of this work in ACP and I would like the authors to either expand the analysis and interpretation significantly before resubmission or consider submission as a technical report.

We have followed the reviewer's suggestion and extended the analysis and interpretation of the results of our study. Please note that the depth of this analysis is limited by the lack of measurements of the chemical composition of $PM_{0.1}$ and the related measurement-based source apportionment studies. This limits our ability to reach firm conclusions about what the model gets right and where it fails. For a lot of the aspects of $PM_{0.1}$ behavior (e.g., composition and sources) our work presents our present understanding based on model predictions (emissions and atmospheric processes) to motivate and help in the design of future studies. The simulated episodes were selected because the model including its meteorological and emissions inventory have been extensively evaluated (and corrected) for these periods. In the revised manuscript, we expanded the discussion addressing the reviewer's comments. Our responses (in black) follow each of the reviewer's comments (in blue).

**(2)** Lines 119-121: More needs to be said about why episodes from 13 and 16 years ago were simulated and used for evaluation. How would the findings from this work change for the present (i.e., 2025) given significant changes in the sources and processes linked to UFP over the past decade?

We selected these years for simulation due to the extensive evaluation of the model for these periods. Several studies (Skyllakou et al., 2014; Patoulias et al., 2018; Patoulias and Pandis, 2022) have used (and improved) the emissions and meteorology for these periods and PMCAMx-UF has shown good performance in reproducing $PM_{2.5}$ levels, composition and sources, as well as ultrafine number concentrations demonstrated high confidence in their evaluation. More recent emission inventories (especially for ultrafine particles) require further evaluation before they can be used with the same level of confidence. Our future work will focus on more recent years, providing a more detailed analysis not only of total concentration but also of the contribution of individual sources. We have now clarified these reasons for the selection of the simulated periods in the revised manuscript.

**(3)** Lines 122: This seems like a much-dated version of WRF, which has now moved to 4.6 and beyond. At a minimum, the key WRF updates between the version used here and the latest version need to be discussed.

The meteorological fields for these episodes were prepared a decade ago using the corresponding version of WRF. The corresponding meteorological fields had been evaluated and were judged to be of good quality for use as inputs in a regional CTM. While the latest version of WRF offers improvements in model physics, computational efficiency, grid flexibility, and data assimilation capabilities, we believe that for the first application and evaluation of PMCAMx-UF for $PM_{0.1}$, it is more appropriate to use meteorological inputs that have already been used and evaluated. This explanation has been added to the revised paper.

**(4)** Section 2 needs to have a separate paragraph describing and discussing the various processes that are likely to strongly influence $PM_{0.1}$ concentrations, size distribution, and composition. It should be made clear where the model is reasonably accurate and where some of the outstanding uncertainties lie.

We have followed the reviewer's suggestion and added a paragraph discussing the processes (nucleation, condensation of low-volatility vapors, coagulation, primary emissions, removal) affecting the $PM_{0.1}$ levels. We have also added some discussion about corresponding uncertainties and potential surprises (e.g. significance for $PM_{0.1}$ of minor sources of $PM_{2.5}$ like natural gas combustion). Given the novelty of the study there is not too much to say about where the model is reasonably accurate. This is one of the objectives of the study.

**(5)** Model-measurement comparison for UPF mass: There are uncertainties built into both the modeled (e.g., numerical diffusion) and measured (e.g., shape factor, UPF density) concentrations of UPF mass/volume. These need to be considered during the evaluation. I understand why 100 nm is considered but I feel that this threshold is arbitrary. Ideally, like $PM_{2.5}$, a cutoff size based on the low point in the 'trough' between the Aitken and accumulation models would be better suited.

We have added a paragraph discussing the uncertainties of the measurements and the model. The 100 nm threshold is somewhat arbitrary, but it has been selected to maintain consistency with previous studies and the used definitions. For example, the U.S. EPA (2025) defines ultrafine particles as those with diameters smaller than 0.1 μm. Given that for $PM_{0.1}$ we are interested in the mass distribution the trough in the number distribution is probably not appropriate. The mass distribution in general increases monotonically in this size range so there is no obvious choice for the cutoff. This point is discussed in the revised paper.

**(6)** Sections 4.1 and 4.2: The results from this study need to be placed in context of other PM fractions (e.g., $PM_1$ and $PM_{2.5}$ concentrations and composition) (some of which have been done in Figures S5 and S6) and compared with past work, with the literature review for studies from CA and US as a starting point. The concentration correlations and compositional differences (or similarities) with other PM fractions will provide answers on why and if smaller particles might be more important for health endpoints.

We have added a paragraph discussing the correlations between $PM_{0.1}$ and $PM_{2.5}$ based on previous studies, and how these may relate to health impacts.

**(7)** PM$_1$ and PV$_{0.1}$: The model predictions for these and comparisons against measurements are perhaps the most important output of this paper. While Section 4.3 offers a qualitative description of the comparisons, I found it lacked depth in interpretation and insight. Many of my questions remained unanswered.

We have expanded the discussion of this section adding analysis for the following questions.

**(8)** Why are the time series data so variable and how does it compare with that for, say, PM$_{2.5}$?

Variability is related to different emission sources in each location in combination with the variation of the meteorology. A discussion of this point has been added.

**(9)** What can we learn about the emissions, transport, and physical/chemical pathways for PM$_{0.1}$ and PV$_{0.1}$?

We have added a discussion of the primary and secondary contributions to PM$_{0.1}$ in the different locations. We also summarize our conclusions about the emissions and chemical pathways that play a major role.

**(10)** In what way are these source terms different between <0.1 and <2.5 µm

We have added a discussion of the predicted differences between the two size modes. Unfortunately, there are no observation-based PM$_{0.1}$ source contribution studies, which does not allow us to evaluate the predicted PM$_{0.1}$ source contributions. This is a topic for future work.

**(11)** What causes the spatial and seasonal variability in model performance?

Emissions are the main driver of spatial and seasonal variability, as they fluctuate on monthly, daily, and hourly timescales. Additionally, emissions are spatially distributed across the simulated domain based on land-use data, contributing to variability from site to site. Similarly, meteorology also plays a crucial role in this variability. Changes in wind patterns, boundary layer height, and temperature inversions are among the key meteorological factors influencing these variations. Finally, photochemistry, since a significant fraction of the predicted PM$_{0.1}$ is secondary, is a major driver of variability. We have now included this point in the revised manuscript.

**(12)** Are there ways in which the model predictions can be presented that allow for more information to be extracted (e.g., if <0.1 µm particles are tied to anthropogenic combustion sources, would presenting the data as a ratio to EC or CO make more sense)?

We have followed the suggestion of the reviewer and tried different ways to present the model predictions. We have now added, for some sites with high EC contribution to PM$_{0.1}$, plots presenting the relationship between predicted PM$_{0.1}$ and EC during wintertime. The corresponding discussion and analysis have been added to the paper.

**(13)** How do the raw size distributions compare between predictions and observations (missed opportunity for a more thorough comparison) and what do they say about aerosol microphysical processes (e.g., nucleation, growth)?

We have added a comparison of the predicted and measured volume distributions in selected sites. Please note that nucleation is a major contributor to the concentration of particle number, but it is a minor and often negligible source of PM$_{0.1}$ mass. Also,

growth is a source of $PM_{0.1}$ mass, but it is also a sink of $PM_{0.1}$ because particles grow larger than 100 nm. This discussion has been added to the paper.

*Specific comments*

**(14)** Line 36: how is specific surface area defined and what are the units?
We now explain that specific surface area is defined as the total surface area of the particles per unit mass and give the corresponding units.

**(15)** Line 46: Always expand the first use of abbreviations (i.e., UCD-P here). Please review this for other abbreviations throughout the manuscript.
We have carefully reviewed all abbreviations in the manuscript and made the corresponding additions.

**(16)** Section 2: How many size bins were used to represent the aerosol size distribution?
We have used 41 size bins. We now mentioned this in the revised manuscript.

**(17)** Line 135: A better approach, documented in Murphy et al. (ACP, 2023) and references therein, is to tie the IVOC emission rate to the total VOC/NMOG instead of POA.
Recent studies (Murphy et al., 2023; Manavi and Pandis, 2022) showed that using POA to estimate IVOC emissions leads to an underprediction of secondary organic aerosol concentrations. In the present work, IVOCs were scaled with POA based on the available, previously evaluated emissions, but future studies will adopt the updated approach. This is now mentioned in the paper.

**(18)** Line 169-170: A useful way to average would be to only consider continental regions.
We have added discussion of the average values for continental regions in the revised paper.

**(19)** Line 199: SOA was only 2-10%. This isn't true during the summer and the text does not make it clear if you are talking about summer or winter results. This comment applies elsewhere in the manuscript as well.
We have corrected the typo and revised the text following the reviewer's recommendation.

**(20)** Scatter plots corresponding to Tables 1 and 2 could be useful to assess the overall performance across cities as well as the model skill in reproducing $PV_{0.1}$.
Scatter plots corresponding to these tables have been added to the SI of the revised manuscript, following the reviewer's suggestion.